# Factors Associated with General Health Screening Participation among Married Immigrant Women in Korea

**DOI:** 10.3390/ijerph16203971

**Published:** 2019-10-18

**Authors:** Jinhee Jeong, Yunhee Lee, Sung Hee Kwon, Jun-Pyo Myong

**Affiliations:** 1Department of medical benefit, National Health Insurance Company, 32, Geongang-ro, Wonju-si 26464, Gangwon-do, Korea; jin87love@naver.com; 2Department of Occupational & Environmental Medicine, Seoul St. Mary’s Hospital, College of Medicine, The Catholic University of Korea, 222, Banpo-daero, Seocho-gu, Seoul 06591, Korea; eyh900@naver.com (Y.L.); amykwon228@catholic.ac.kr (S.H.K.)

**Keywords:** married immigrant women, health screening, healthcare disparities, ethnicity

## Abstract

*Background*: The number of married female immigrants living in Korea has been increasing and is expected to increase further. This study was performed to identify factors associated with national general health screening participation among married immigrant women living in South Korea. *Methods*: The Korean National Health Insurance System’s (NHIS) customized database for the years 2014 and 2015 was used. The targets of this study were women aged 19 years old and above. To identify factors associated with national general health screening participation, the following analyses were employed: frequency, chi-square, simple regression, and multiple regression. *Results*: A total of 11,213 women were identified in the NHIS database. Overall, 67.4% participated in national general health screenings, lower than the 74.6% participation rate of the entire women’s health screening program. Married immigrant women with a job had higher health screening participation than those without a job (OR = 2.822, *p* < 0.0001). Age, socioeconomic status, and duration of stay were related to health screening behaviors among employed married immigrant women. Nationality, socioeconomic status, duration of stay, and disease status were associated with general health screening behaviors among unemployed immigrant women. The odds ratios decreased as the length of stay increased, regardless of employment status. *Conclusion*: The results of this study showed that employment status and duration of stay in Korea are significantly associated with general health screening participation. Accordingly, to improve awareness about health screening and health care disparities, programs promoting health screening participation for socially vulnerable classes, including immigrant women and unemployed women, should be instigated.

## 1. Introduction

General health screening aims to improve public health and reduce medical expenses by the early detection of cardiovascular diseases and conditions such as hypertension, diabetes mellitus, and hyperglycemia, and by connecting participants with treatment and management [1,2]. In accordance with Article 52 of the National Health Insurance Act, the Korean National Health Insurance Service (NHIS) provides general health screenings for the early detection of diseases to subscribers and dependents, as well as medical care benefits [3].

Health screening increases participants’ interest in health because health information can be accessed during health screenings. Screenings are effective in encouraging participants to practice healthy lifestyles including eating an appropriate diet and regular exercise [4]. National health screening reduces death by the early detection of diabetic patients, thus positively affecting long-term health [5,6,7]. The better the general health screenings, the lower the number of serious diseases that develop, and medical expenses have also been reduced [6,8]. Health screenings are also associated with reductions in cerebral cardiovascular disease and average hospitalization days [7,9,10].

According to the population survey of the Statistics Office of Korea, the number of marriages with foreigners was 20,591 in 2016, which represents 7.3% of total marriages (281,600). Among these, there were 14,800 marriages between foreign women and Korean men [11]. The number of married female immigrants living in Korea has been increasing [12] and is expected to rise further because of the specific structure of the Korean population. More attention and effort are needed at the national level to help married immigrant women who are struggling to adjust to South Korean society.

In addition, married immigrant women have a high dependency on family members due to their migrant situation, which can be an obstacle to adopting appropriate health behavior including the use of health services. Most married immigrant women have migrated from underdeveloped countries and may be in poor health. Compared with Korean women, the income level of women from multicultural families is more likely to be lower than the middle and lower national income levels, which represents an additional barrier to these women in managing their health [13,14,15].

Therefore, this study was performed to identify the factors associated with general health screening behaviors among married immigrant women living in South Korea.

## 2. Materials and Methods

### 2.1. Study Population and Source

The NHIS’s customized dataset was used for this study. Two annual health screening databases for the years 2014 and 2015 were used to check the status of participants in general health screenings in South Korea. To identify factors related to general screening behaviors, qualifications and medical history data were collected. Information on women who acquired a marriage immigrant visa (F-6) was derived from the NHIS customized dataset. The marriage immigration status visa (F-6) is issued to foreigners who have married Koreans. It is also issued to a person whom the Minister of Justice recognizes as raising a child born in a Korean marital relationship or to a person who cannot maintain a normal marital relationship for reasons such as the death or disappearance of a Korean spouse.

In total, 49,786 women with an F-6 visa who were using the NHIS during the study period (2014–2015) were enrolled in the present study. Exclusion criteria were as follows: ineligibility for general health screenings by Korean NHIS scheme (*n* = 38,373) (women over 40 years old and unemployed women at the general health screening year, or women under 40 years old & unemployed women), women born before 1940 (*n* = 36), women with missing data on residency and insurance contribution (*n* = 102), and women with an invalid entry date to Korea (*n* = 62). A total of 11,213 female immigrants aged 19 and over were eligible for the present study. The Institutional Review Board of the Seoul St. Mary’s Hospital approved this study (KC17ZES10589) before the researchers accessed the NHIS data.

### 2.2. Definition of Variables

Age was classified into ≤39 years, 40–59 years, and ≥60 years. The nations of origin were categorized as follows: China, Vietnam, the Philippines, Japan, Mongolia, Thailand, and other countries (69 countries). Residential areas are categorized as Seoul, metropolitan cities (Busan, Daegu, Incheon, Gwangju, Daejeon, Ulsan, and Sejong), and rural areas (Gyeonggi-do, Gangwon-do, Chungcheong-do, Jeolla-do, Gyeongsang-do, and Jeju-do). The NHIS extracts the payment cost from the subscriber’s total household salary, house cost, type of cars, etc. Therefore, the health insurance contributions reflect the socioeconomic status of subscribers. As an indicator of socioeconomic levels, 20 grades of health insurance contributions are used. Q1 for grades 1–5 (the lowest income classes), Q2 for grades 6–10, Q3 for grades 11–15, and Q4 for grade 16–20 (the highest income classes). The period of stay was set as the difference between the date of their general health screenings (for non-participants, the last day of each year) and their first entry date to Korea, and was grouped as follows: <5, 5–9, 10–14, and ≥15 years. According to the Charlson Comorbidity Index based on the International Classification of Diseases (ICD) diagnosis codes, there are 10 kinds of diseases (dementia, connective tissue disease, ulcer disease, myocardial infarction, congestive heart failure, chronic pulmonary disease, peripheral vascular disease, cerebrovascular disease, diabetes mellitus, and mild liver disease) that are taken into consideration. Participants were classified into a group with diseases (one or more) and a group without diseases. Job status was classified by employed versus unemployed. Participation in the general health screening was a dependent variable, and was classified as persons who participated in the general health screening and non-participants.

### 2.3. Statistical Analysis

All data collected in this study were analyzed using SAS version 9.4 (SAS Institute Inc., Cary, NC, USA). The general characteristics of the subjects were identified by frequency, mean, and standard deviation. Student *t* test was performed for continuous variables. A *chi-square* test was performed to identify general health screening behavior according to the general characteristics. A simple logistic regression was performed to analyze the association between age, nationality, residential area, socioeconomic level, duration of stay, disease, employment status and general health screening behaviors. Multiple logistic regression analysis was performed to identify factors related to general health screening behaviors according to employment status.

## 3. Results

### 3.1. Subject Characteristics

The general characteristics of the subjects are shown in Table 1. The average age of participants was 43.6 ± 10.5 years. The number of women from China was the highest at 6996 (62.4%). The number of residents living in provincial areas was 6538 (58.3%). The insurance contribution income quartile Q2 was the highest at 4246 (37.9%) participants. The average length of stay in Korea was 6.8 ± 3.8 years. Overall, 4520 (40.3%) women had one or more diseases, with ulcers being the most frequent (*n* = 1933). A total of 22.5% (*n* = 2522) of participants were employed.

### 3.2. Health Screening Rates

The participation rate for general health screening was 67.4% (*n* = 7559) of married female immigrants. There were significant differences in participation rates for general health screenings according to age, nationality, residential area, socioeconomic level, duration of stay, diseases, and employment status (Table 1). Married immigrant females from China and Vietnam, recently immigrated, with co-morbidity, and employed were more likely to participate in General Health Examinations.

The general health screening rates of unemployed and employed Chinese women were 66.3% and 88.9%, respectively. After stratification for employment status, the statistically significant difference in the general screening rate of married female immigrants according to nationality disappeared among employed women. There was no significant difference in screening participation depending on disease status among employed females. In unemployed women, the general health screening participation for women with diseases was 69.5%, which is higher than the 57.9% for women without diseases (*p* < 0.01) (Table 2).

### 3.3. Factors Associated with Health Screening

Table 3 shows the results of logistic regression analysis of the factors associated with the national general health screening participation among married immigrant females in Korea in 2014 and 2015. In Model I, those with older age (≥60), Chinese nationality, Q2 and Q3 insurance fee status, relatively recent immigration, and co-morbidities, and those with employment showed a positive association with participation in General Health Examinations. After stratification by employment status from Model I (Model II as unemployed/employed), women 60 years or older, who were Chinese, in the Q3 insurance fee bracket, were recently immigrated, and those with co-morbidities were associated with participation in General Health Examinations among unemployed, married immigrant women (*p* < 0.05). However, the statistically significant associations between participation in General Health Examination, co-morbidities and nationality were not observed among employed in Model II.

## 4. Discussion

This study was performed in order to identify factors associated with national general health screening participation for married immigrant women living in South Korea. Age, socioeconomic level and duration of stay were associated with participation in the General Health Examination. Employment status was a potential compounding factor for the association between participation in General Health Examinations and the characteristics of married immigrant females (co-morbidity and nationality) in South Korea.

According to the annual statistics of health screenings in 2015, the participation rate of married immigrant females in health screenings was 67.4%, which was lower than that of 74.6% for all eligible females [1]. This study showed that 36.9% of married immigrant females in Korea belonged to the lowest insurance fee quartile (lowest socioeconomic status). Previous literature on healthcare accessibility inequality showed that low socioeconomic status populations have a lower utilization of healthcare programs [16,17]. This could be due to low socioeconomic status populations having less time to participate in general health screening programs, even when these are cost free [18]. In the current study, unemployed married immigrant females in the lowest quartile showed the lowest participation rate for general health examinations. However, this vulnerability was not seen in employed participants. The odds ratio for participation in General Health Examinations among participants in insurance fee quartiles Q1–Q3, was higher than those for the reference group (Q4, the highest socioeconomic status group). This suggests that employment status affects participation in General Health Examinations regardless of salary.

The health screening participation rate of employed females was significantly higher than that of unemployed females. This is because employers conduct health screenings to protect employees’ health. The Ministry of Employment and Labor encourages health screening participation at workplaces and can impose a fine in accordance with Article 72 of the Industrial Safety and Health Act if this is not provided [19]. The current results are consistent with a previous study that the health screening rate of employed women is higher than that of unemployed women [20]. To improve health screening participation, it will be necessary to carry out a special program for married immigrant women who are unemployed, and further research is needed to analyze the factors that prevent female immigrants from being screened.

For married immigrant women with an unemployed status, the general health screening participation of female immigrants with disease was significantly higher than that of married immigrant women without disease. At the age of 40 and over, the possibility of disease increases. As females with diseases are more likely to get information about their health status and recognize the need for healthcare during hospital visits, their interest in healthcare increases. There is a high possibility that medical staff will encourage patients who present at hospitals to participate in health screenings. Therefore, the screening rate among those with co-morbidities are considered to be high for these reasons. In a study using multicultural family survey data, subjects with a disease for 2 weeks had a higher uptake of health screening than those who did not have a disease [21]. Women with one or more diseases have higher rates of screening participation than women without diseases [22], and this result is consistent with the results of this study. For unemployed married immigrant females, healthcare providers such as physicians or nurses in hospitals or public health centers should advise them to utilize general health examination services.

As there was a significant difference in health screening behaviors according to nationality in unemployed women, it is necessary to carry out health screening promotion projects that take into consideration the social and cultural characteristics of each country. The formation of diverse networks of Chinese female immigrants in Korea, compared to the relatively small number of networks of women from other countries [23], encourages participation in health screenings. In addition, Chinese female immigrants in Korea have higher levels of support from their families, such as from their husbands and relatives in Korea, than females from other countries [24]. Social support is known to have a positive effect on health promotion behavior [25], and this is believed to affect health screening participation. However, there was no statistically significant difference in participation in health screening by nationality among employed females because employed females were influenced by company health screenings rather than by nationality.

This study showed that regardless of employment status, the longer the duration of stay in Korea, the lower the health screening participation. For married immigrant females, income level might be higher among those who stay longer in Korea [26]. Those with high income might choose more expensive health screening programs that perform more diverse examinations than those available with national health screening programs [27]. Therefore, the longer duration of stay in Korea might be a possible association with a higher salary which is concordant with this analysis (*p* for trend <0.01 data was not shown in the table); consequently, married immigrant females might participate in a health screening privately instead of a national health screening. In order to increase the health screening rate of long-term stay immigrant females in Korea, continuous health screening promotion projects should be instigated.

There are several limitations to this study. Firstly, there was a limitation in selecting variables because data such as qualifications and disease details were collected using the NHIS customized database. Various variables such as educational levels, social support, and living satisfaction that could affect health status, were not considered in this study. However, factors that could not be confirmed by simple secondary data such as duration of stay and nationality were identified. Since the subjects were females with marriage visas at the time of screening participation, the study did not take into consideration females who obtained Korean citizenship or changed visas due to divorce or the acquisition of permanent residency. Secondly, there might be potential information bias on definition of employment status (employed vs. (unemployed or not employed)). Those unemployed women would not represent those looking for work due to the mixture of the retired or not employed. Therefore, an information bias should be considered to interpret the present study. Further study should be conducted in order to evaluate the effect of confounding.

Despite these limitations, the study was effective in analyzing the factors associated with a general health screening for married immigrant females living in Korea. This is the first research on general health examinations conducted on married immigrant females across Korea.

## 5. Conclusions

In conclusion, age, duration of stay, socioeconomic status, and employment status are associated with participation in General Health Examinations among married immigrant females in Korea. However, further research on health screening behaviors and medical service use is required to address the current lack of information on health screening behaviors of married female immigrants living in Korea, in order to promote the health of those who have become important members of our society.

## Figures and Tables

**Table 1 ijerph-16-03971-t001:** General characteristics of eligible participants.

Variables	Categories	Mean ± SD *N* (%)	General Health Screening		*p*
			No	Yes	
Age (years)		*43.6 ± 10.5*	*43.8 ± 9.8*	*43.6 ± 10.8*	0.3053
	<40	2987(26.6)	886(29.7)	2101(70.3)	0.0019
	40–59	7559(67.4)	2556(33.8)	5003(66.2)	
	≥60	667(6.0)	212(31.8)	455(68.2)	
Nationality	China	6996(62.4)	2201(31.5)	4795(68.5)	<0.0001
	Vietnam	1888(16.8)	486(25.7)	1402(74.3)	
	Philippines	558(5.0)	199(35.7)	359(64.3)	
	Japan	664(5.9)	282(42.5)	382(57.5)	
	Mongolia	214(1.9)	95(44.4)	119(55.6)	
	Thailand	210(1.9)	104(49.5)	106(50.5)	
	Others	683(6.1)	287(42.0)	396(58.0)	
Residence	Seoul	2576(23.0)	898(34.9)	1678(65.1)	0.0187
	Metropolitan	2099(18.7)	664(31.6)	1435(68.4)	
	Province	6538(58.3)	2092(32.0)	4446(68.0)	
Insurance fee quartile	Q1	4136(36.9)	1470(35.5)	2666(64.5)	0.0420
	Q2	4246(37.9)	1269(29.9)	2977(70.1)	
	Q3	2048(18.3)	622(30.4)	1426(69.6)	
	Q4	783(7.0)	293(37.4)	490(62.6)	
Duration of stay (years)		*6.8 ± 3.8*	*7.7 ± 4.0*	*6.3 ± 3.6*	<0.0001
	<5	3858(34.4)	913(23.7)	2945(76.3)	<0.0001
	5–9	5579(49.8)	1969(35.3)	3610(64.7)	
	10–14	1382(12.3)	565(40.9)	817(59.1)	
	≥15	394(3.5)	207(52.5)	187(47.5)	
Co-morbidity *	No	6693(59.7)	2388(35.7)	4305(64.3)	<0.0001
	Yes	4520(40.3)	1266(28.0)	3254(72.0)	
Employment status	No	8691(77.5)	3219(37.0)	5472(63.0)	<0.0001
	Yes	2522(22.5)	435(17.3)	2087(82.7)	
Total		11,213(100.0)	3654(32.6)	7559(67.4)	

Italics: Student *t* test was performed. *: subjects with one or more co-morbid diseases according to the Charlson Comorbidity Index (dementia, connective tissue disease, ulcer disease, myocardial infarction, congestive heart failure, chronic pulmonary disease, peripheral vascular disease, cerebrovascular disease, diabetes mellitus, and mild liver disease).

**Table 2 ijerph-16-03971-t002:** Participation in General Health Examinations among married immigrant females in Korea by employment status.

Variables	Categories	Employment Status					
		Unemployed			Employed		
		Participation of GHE		*p* Value	Participation of GHE		*p* Value
		Yes	No		Yes	No	
Age(years)	*Mean ± SD*	*46.7 ± 9.4*	*45.1 ± 9.1*	<0.0001	*35.3 ± 10.0*	*33.6 ± 8.6*	<0.0001
	<40	706(56.2)	550(43.8)	<0.0001	1395(80.6)	336(19.4)	0.0001
	40–59	4335(63.8)	2460(36.2)		668(87.4)	96(12.6)	
	≥60	431(67.3)	209(32.7)		24(88.9)	3(11.1)	
Nationality	China	4042(66.3)	2057(33.7)	<0.0001	753(83.9)	144(16.1)	0.2380
	Others	1430(55.2)	1162(44.8)		1334(82.1)	291(17.9)	
District	Seoul	1394(63.1)	815(36.9)	0.9849	284(77.4)	83(22.6)	0.0115
	Metropolitan	981(62.8)	580(37.2)		454(84.4)	84(15.6)	
	Province	3097(62.9)	1824(37.1)		1349(83.4)	268(16.6)	
Insurance fee quartile	Q1	1539(56.3)	1193(43.7)	<0.0001	1127(80.3)	277(19.7)	0.0611
	Q2	2127(65.0)	1145(35.0)		850(87.3)	124(12.7)	
	Q3	1324(69.0)	594(31.0)		102(78.5)	28(21.5)	
	Q4	482(62.7)	287(37.3)		8(57.1)	6(42.9)	
Duration of stay (years)	*Mean ± SD*	*6.8 ± 3.6*	*7.9 ± 4.1*	<0.0001	*5.0 ± 3.1*	*5.8 ± 8.8*	<0.0001
	<5	1805(71.7)	714(28.3)	<0.0001	1140(85.1)	199(14.9)	0.0007
	5–9	2788(61.1)	1772(38.9)		822(80.7)	197(19.3)	
	10–14	707(56.9)	535(43.1)		110(78.6)	30(21.4)	
	≥15	172(46.5)	198(53.5)		15(62.5)	9(37.5)	
Co-morbidity *	No	2833(57.9)	2063(42.1)	<0.0001	1472(81.9)	325(18.1)	0.0797
	Yes	2639(69.5)	1156(30.5)		615(84.8)	110(15.2)	
Total		5472(63.0)	3219(37.0)		2087(82.8)	435(17.2)	<0.0001

Italics: Student *t* test was performed. *: subjects with one or more co-morbid diseases according to the Charlson Comorbidity Index (dementia, connective tissue disease, ulcer disease, myocardial infarction, congestive heart failure, chronic pulmonary disease, peripheral vascular disease, cerebrovascular disease, diabetes mellitus, and mild liver disease).

**Table 3 ijerph-16-03971-t003:** The logistic regression analysis of health screening participation.

Variables	Categories	Crude Analysis	Multiple Analysis		
			Model I ^§^	Model II ^‖^	
		OR	OR	Unemployed OR	Employed OR
Age (years)	<40	reference	reference	reference	reference
	40–59	0.825 *	1.262	1.130	2.151
	≥60	0.905	1.474 *	1.301 *	2.677
Nationality	China	reference	reference	reference	reference
	Others	0.873 *	0.768 ^‡^	0.695 ^‡^	1.250
District	Seoul	reference	reference	reference	reference
	Metropolitan	1.157	1.169	1.098	1.689 *
	Province	1.137	1.138	1.083	1.530
Insurance fee quartile	Q1	1.084 *	0.723 ^‡^	0.692 ^‡^	2.125
	Q2	1.403 ^‡^	1.093 ^*^	1.033	3.759 ^†^
	Q3	1.371 *	1.278 ^‡^	1.286 ^‡^	2.078
	Q4	reference	reference	reference	reference
Duration of stay (years)	<5	reference	reference	reference	reference
	5–9	0.568 *	0.605	0.589	0.678 *
	10–14	0.448 *	0.504 *	0.499 *	0.540
	≥15	0.280 ^‡^	0.356 ^‡^	0.386 ^‡^	0.165 *
Co-morbidity **	No	reference	reference	reference	reference
	Yes	1.426 ^‡^	1.563 ^‡^	1.625 ^‡^	1.172
Employment status	No	reference	reference		
	Yes	2.822 ^‡^	3.737 ^‡^		

* *p* < 0.05, † *p* < 0.001, ‡ *p* < 0.0001. § Model I: age, nationality, district, insurance fee, duration of stay, disease and employment status are adjusted; ‖ Model II: age, nationality, district, insurance fee, duration of stay and disease are adjusted; **: subjects with one or more co-morbid diseases according to the Charlson Comorbidity Index (dementia, connective tissue disease, ulcer disease, myocardial infarction, congestive heart failure, chronic pulmonary disease, peripheral vascular disease, cerebrovascular disease, diabetes mellitus, and mild liver disease).

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
