# Peer review of "Factors Associated with General Health Screening Participation among Married Immigrant Women in Korea"

_ijerph, 2019, doi:10.3390/ijerph16203971_

Round 1

Reviewer 1 Report

This study uses high quality administrative data to analyse factors associated with participation of female immigrants in Korea.

Focus on “married immigrant women”

I realise from the reference list that in Korea it is very common for studies to look specifically at married immigrant women. Is a visa type F-6 for marriage the most common entry path for female immigrants to Korea? In the context of this research question the restriction to married women seems rather arbitrary, therefore it would be good to know if this simply reflects most women’s circumstances.

This probably also means that there most (female) immigrants in Korea automatically have a Korean person (their spouse) in their social network for support. This would be quite different to many immigrant–receiving Western countries, such as the US, Canada or many European countries, where female immigrants are often married to co-nationals.

Line 72: Age 40

On the one hand, one of the exclusion criteria is ineligibility for general health screenings (mainly unemployed or women under 40 years old). On the other hand, you use “under 40 years old” as the reference category, and in the sample 27% of women are in this age group

Line 83 SEP / Insurance class

For readers not familiar with the Korean health insurance system - is the income the joint income of the couple, or only the woman’s income? Given that unemployed women also have a value for insurance quartile, I assume it is the joint income?

Line 92 Employment status

You distinguish between ‘employed’ and ‘unemployed’. As there is no other category, am I right in thinking that the ‘unemployed’ category includes women who are retired or staying at home looking after the house? In this case, I think it would be better to call this group ‘not employed/not working’, as ‘unemployed’ would suggest women are looking for work.

I also don’t understand why 78% of the sample are ‘unemployed’ women when being unemployed is one of the exclusion criteria (though this might be logical, if in the exclusion criteria unemployed actually means only ‘job seeking’, not anyone not working).

Section 3.2 – I am struggling slightly to link the text to the relevant tables:

Line 114 “There were significant differences in participation rates…(Table 2)”: Should this be table 1 instead of table 2? Because table 1 shows the chi square tests for the whole sample, while table 2 presents results already stratified by employment status. Also the next sentence “Married immigrant females… employed were more likely to participate..” can only refer to table 1?

Line 120 “After stratification for employment status… difference according to nationality disappeared”. If I read table 2 correctly it only disappeared amongst employed women, but not amongst not employed women?

Section 3.3

Line 131 “After stratification…” this sentence seems to refer to the model for unemployed women?

Then the next sentence continues “However, the.. associations between participation…. and co-morbidities and nationality was not observed after stratification..”. This contradicts the previous sentence but I think that’s because this sentence summarises the ‘employed’-only model?

Table 2: In the total row the totals for the “no” columns are missing.

Table 3: The regression models should have case numbers added.

Discussion Line 170 “This suggests…”

I don’t think the results of the stratified models 2 (unemployed vs employed) can be interpreted in a way where one can make a statement regarding the effect of the stratification variable (employment status) on the outcome. In order to test a statement such as “employment status affects participation in General Health Examinations regardless of salary” one would need to use model 1 and add an interaction term between employment status and insurance quartile. If the interaction term were not significant this would support this statement.

In my opinion, from Model 2 (employed) one could only say that amongst employed women insurance quartile (=salary) does not predict screening participation. However, even this is not clearly supported as women in Q2 are much more likely to participate than the women in the reference quartile.

Author Response

Reviewer #1

Comment #1 Age 40. On the one hand, one of the exclusion criteria is ineligibility for general health screenings (mainly unemployed or women under 40 years old). On the other hand, you use “under 40 years old” as the reference category, and in the sample 27% of women are in this age group

Reply #1) Thanks for your comment. The eligibility of general health screening from Korean NHIS are those workers at any age over 15 or older and 40 years or older among not working population. Therefore, the authors want to avoid the eligibility of general health screening, the definition of ineligibility was re-described as following;

“women over 40 years old and unemployed women at the general health screening year, or women under 40 years old & unemployed women”

Comment #2 SEP / Insurance class. For readers not familiar with the Korean health insurance system - is the income the joint income of the couple, or only the woman’s income? Given that unemployed women also have a value for insurance quartile, I assume it is the joint income?

Reply #2) As your sincere recommendation, the authors describe how the NHIS calculate the insurance payment class as following;

“The NHIS extracts the payment cost from subscriber total household’s salary, house cost, type of cars and etc.. Therefore, the health insurance contributions reflect the socioeconomic status of subscribers.”

Comment #3 Employment status. You distinguish between ‘employed’ and ‘unemployed’. As there is no other category, am I right in thinking that the ‘unemployed’ category includes women who are retired or staying at home looking after the house? In this case, I think it would be better to call this group ‘not employed/not working’, as ‘unemployed’ would suggest women are looking for work.

Reply #3) Thank you for concise comment on the present article. The authors described those potential information bias as unemployed and not employed women in the discussion section as following;

“Those unemployed women would not represent those who looking for work due to mixture of retired or not employed. Therefore, an information bias should be considered to interpret the present study, further next study should be followed to evaluate the effect of confounding.”

Comment #4 1) Line 114 “There were significant differences in participation rates…(Table 2)”: Should this be table 1 instead of table 2? Because table 1 shows the chi square tests for the whole sample, while table 2 presents results already stratified by employment status. Also the next sentence “Married immigrant females… employed were more likely to participate..” can only refer to table 1?

2) Line 120 “After stratification for employment status… difference according to nationality disappeared”. If I read table 2 correctly it only disappeared amongst employed women, but not amongst not employed women?

Reply #4) Thank you for your sincerely comment on the present article.

The table 1 is right so we corrected the number of table from table 2 à table 1. which is highlighted in the manuscript. The authors updated the manuscript according to indicated comment as you said as following:

“After stratification for employment status the statistically significant difference in the general screening rate of married female immigrants according to nationality disappeared among employed women.”

Comment #5 Line 131 “After stratification…” this sentence seems to refer to the model for unemployed women? Then the next sentence continues “However, the.. associations between participation…. and co-morbidities and nationality was not observed after stratification..”. This contradicts the previous sentence but I think that’s because this sentence summarises the ‘employed’-only model?

Reply #5) Thank you for your sincerely comment on the present article. The authors specified the meaning of table 3 as following;

“Table 3 shows the results of logistic regression analysis of the factors associated with the national general health screening participation among married immigrant females in Korea in 2014 and 2015. In Model I, those with older age (≧60) Chinese nationality, Q2 & Q3 insurance fee status, relatively recent immigration, and co-morbidities, and those with employment showed a positive association with participation in General Health Examinations. After stratification by employment status from Model I (Model II as unemployed/employed), women 60 years or older, who were Chinese, in the Q3 insurance fee bracket, were recently immigrated, and those with co-morbidities were associated with participation in General Health Examinations among unemployed married immigrant women (p<0.05). However, the statistically significant associations between participation in General Health Examination and co-morbidities and nationality was not observed among employed in Model II.”

Comment #6 Table 2: In the total row the totals for the “no” columns are missing.

Reply #6) Thank you for your sincerely comment on the present article. The authors updated as you commented in table 2. In addition, the updated contents were highlighted in Table 2.

Comment #7 Table 3: The regression models should have case numbers added

Reply #7) Thank you for your sincerely comment on the present article. The authors considered the comment carefully. After the discussion on the numbers of logistic regression model, the numbers what you requested was shown in table 1 and table 2 therefore, the authors did not insert the numbers of screening participated in table 3. This is a concern on the reduce the redundant of table. Please consider our intention.

Comment #8 Discussion Line 170 “This suggests…” I don’t think the results of the stratified models 2 (unemployed vs employed) can be interpreted in a way where one can make a statement regarding the effect of the stratification variable (employment status) on the outcome. In order to test a statement such as “employment status affects participation in General Health Examinations regardless of salary” one would need to use model 1 and add an interaction term between employment status and insurance quartile. If the interaction term were not significant this would support this statement. In my opinion, from Model 2 (employed) one could only say that amongst employed women insurance quartile (=salary) does not predict screening participation. However, even this is not clearly supported as women in Q2 are much more likely to participate than the women in the reference quartile.

Reply #8) Thank you for your sincerely comment on the present article. The authors performed a interaction term in Model 1. The result of interaction term was following;

Analysis of Maximum Likelihood Estimates

Parameter

DF

Estimate

Standard Error

Wald Chi-Square

Pr > ChiSq

Insurance*employment status

1

0.1193

0.1096

1.1853

0.2763

The interaction term result was not statistically significant. Therefore, the authors did not changed the details of discussion.

Reviewer 2 Report

1) Use a consistent comma within Results section. The numbers in lines 108 and 111 have commas, whereas the numbers in lines 107, 109, and 110 do not have commas.

2) In lines 79-80, "other countries (69 countries)" is intriguing as this is a substantial amount of countries.  If the countries are too many to mention, could you possibly mention geographic region or continent (e.g., Asia, Africa, Europe, etc.) to add context to that number of 69 countries?

3) The results in lines 119-120 ("The general health screening rates of unemployed and employed Chinese women were 66.3% and 55.2%, respectively.") are intriguing.  Is there a theory or explanation for why the screening rates were higher among unemployed women?  In lines 120-122, it then says "After stratification for employment status the statistically significant  difference in the general screening rate of married female immigrants according to nationality disappeared."  Does this sentence refer to the results from lines 119-120?  Can you expand upon what the stratification for employment status entailed?

4) In lines 207-208, check the wording of the sentence "Those with high income might chose more..."

5) Overall, this is an interesting, insightful, and relevant paper.

Author Response

Reviewer #2

Comment #1 Use a consistent comma within Results section. The numbers in lines 108 and 111 have commas, whereas the numbers in lines 107, 109, and 110 do not have commas.

Reply #1) The authors inserted commas as following:

“The general characteristics of the subjects are shown in Table 1. The average age of participants was 43.6 ± 10.5 years. The number of women from China was the highest at 6996 (62.4%). The number of residents living in provincial areas was 6,538 (58.3%). Insurance contribution income quartile Q2 was the highest at 4,246 (37.9%) participants. The average length of stay in Korea was 6.8 ± 3.8 years. Overall, 4,520(40.3%) women had one or more diseases, with ulcer being the most frequent (n=1,933). A total of 22.5% (n=2,522) of participants were employed.

3.2. Health screening rates

The participation rate for general health screening was 67.4% (n=7,559) of married female immigrants.”

Comment #2 In lines 79-80, "other countries (69 countries)" is intriguing as this is a substantial amount of countries.  If the countries are too many to mention, could you possibly mention geographic region or continent (e.g., Asia, Africa, Europe, etc.) to add context to that number of 69 countries?

Reply #2) The authors originally considered as reviewer’s comment. However, majority of married immigrants were Asian, so there was a difficulty to show small amount of percents. In addition, we thought that the distribution is not good for showing.

Comment #3 The results in lines 119-120 ("The general health screening rates of unemployed and employed Chinese women were 66.3% and 55.2%, respectively.") are intriguing.  Is there a theory or explanation for why the screening rates were higher among unemployed women?  In lines 120-122, it then says "After stratification for employment status the statistically significant  difference in the general screening rate of married female immigrants according to nationality disappeared."  Does this sentence refer to the results from lines 119-120?  Can you expand upon what the stratification for employment status entailed?

Reply #3) The authors re-described those to avoid the confusing meaning as following;

“Table 3 shows the results of logistic regression analysis of the factors associated with the national general health screening participation among married immigrant females in Korea in 2014 and 2015. In Model I, those with older age (≧60) Chinese nationality, Q2 & Q3 insurance fee status, relatively recent immigration, and co-morbidities, and those with employment showed a positive association with participation in General Health Examinations. After stratification by employment status from Model I (Model II as unemployed/employed), women 60 years or older, who were Chinese, in the Q3 insurance fee bracket, were recently immigrated, and those with co-morbidities were associated with participation in General Health Examinations among unemployed married immigrant women (p<0.05). However, the statistically significant associations between participation in General Health Examination and co-morbidities and nationality was not observed among employed in Model II.”

Comment #4 In lines 207-208, check the wording of the sentence "Those with high income might chose more..."

Reply #4) Thank you for your sincerely comment on the present article. The authors specified the detail description for checking the meaning as following;

Those with high income might chose more expensive health screening programs that perform more diverse examinations than those of national health screening programs Therefore, the longer duration of stay might have more salary so that they might participate a health screening privately not national health screening.

Reviewer 3 Report

Dear authors,

Thank you for your interesting paper regarding the important research field. Nevertheless, I made some comments and suggestions.

Abstract (and Conclusions):

It would be instructive if the the impact of the predicting factors would be more clearly described (e.g. "lower age, lower insurance fee quartile and longer duration of stay is associated with....") than only to state associations.

Keywords:

If there is no specific limit, keywords like migration, ethnicity, inequality or utilization could be added.

Methods:

2.1. Exclusion criteria: It is not quite clear was the criteria "ineligibility for general health screenings" mean? About 38 000 individuals dropped out.

2.2.  The term "socioeconomic status" (in the text) - when in fact insurance fee (as it is in the tables) is assessed - should be handled with care.

2.3. The procedures should be stated more clearly, e.g. "Multiple logistic regression analysis was performed.....with and without stratification for employment status.

Results: 

Table 1: In the line "employment status" , a "No" (instead of an "N") has to be included.

3.2. Page 3, lines 119 to 120: Is the secreening rate 55,2% for employed chinese women or is it 83.9% (Table 2)?

In general, in terms of insurance quartile fee (stated as "SES") there are some associations but not really a social gradient as it is known from the literature. So associations with some single status groups should be interpreted with care.

Table 3: In the table, it is not explained what "simple analysis" mean. An unadjusted model with single regressions? It could also be explained in the methods. Also, confidence intervals would be instructive (but not obligatory).

Author Response

Reviewer #3

Comment #1 Abstract (and Conclusions): It would be instructive if the the impact of the predicting factors would be more clearly described (e.g. "lower age, lower insurance fee quartile and longer duration of stay is associated with....") than only to state associations.

Keywords: If there is no specific limit, keywords like migration, ethnicity, inequality or utilization could be added.

Reply #1) Thank you for comment us for upgrading the present article. For abstract, the authors tried to summarize concisely. If we intend most of results describe to show the detail classifications, the contents will be much which is not proper for the present journal guideline as concise abstract. Please consider this.

For Keywords; the authors inserted the Ethnicity as Keywords, however, the migration, inequality or utilization seems the similar meaning of Married immigrant women; Health screening; Healthcare disparities, therefore, the authors did not insert them.

Comment #2 2.1. Exclusion criteria: It is not quite clear was the criteria "ineligibility for general health screenings" mean? About 38 000 individuals dropped out

Reply #2)

Thanks for your comment. The eligibility of general health screening from Korean NHIS are those workers at any age over 15 or older and 40 years or older among not working population. Therefore, the authors want to avoid the eligibility of general health screening, the definition of ineligibility was re-described as following;

“women over 40 years old and unemployed women at the general health screening year, or women under 40 years old & unemployed women”

Comment #3 The term "socioeconomic status" (in the text) - when in fact insurance fee (as it is in the tables) is assessed - should be handled with care.

Reply #3) As your sincere recommendation, the authors describe how the NHIS calculate the insurance payment class as following;

“The NHIS extracts the payment cost from subscriber total household’s salary, house cost, type of cars and etc.. Therefore, the health insurance contributions reflect the socioeconomic status of subscribers.”

Comment #4 The procedures should be stated more clearly, e.g. "Multiple logistic regression analysis was performed.....with and without stratification for employment status.

Reply #4) Thank you for your sincerely comment on the present article.

Comment #5 Table 1: In the line "employment status" , a "No" (instead of an "N") has to be included.

Reply #5) Thank you for your sincerely comment on the present article. The authors updated as you commented in table 2. In addition, the updated contents were highlighted in Table 2.

Comment #6 Page 3, lines 119 to 120: Is the screening rate 55,2% for employed chinese women or is it 83.9% (Table 2)?.

Reply #6) Thank you for your critical comment on the present article. The authors revised the percent as 88.9%.

“The general health screening rates of unemployed and employed Chinese women were 66.3% and 88.9%, respectively.”

Comment #7 In general, in terms of insurance quartile fee (stated as "SES") there are some associations but not really a social gradient as it is known from the literature. So associations with some single status groups should be interpreted with care.

Reply #7) Thank you for your sincerely comment on the present article. As replied for comment #3, the authors describe how the NHIS calculate the insurance payment class as following;

“The NHIS extracts the payment cost from subscriber total household’s salary, house cost, type of cars and etc.. Therefore, the health insurance contributions reflect the socioeconomic status of subscribers.”

Therefore, the insurance fee reflect the SES of income level of subscribers.

Comment #8 Table 3: In the table, it is not explained what "simple analysis" mean. An unadjusted model with single regressions? It could also be explained in the methods. Also, confidence intervals would be instructive (but not obligatory).

Reply #8) The term of simple analysis was changed into crude analysis in table 3.